# Gynecological Speculums in the Context of the Circular Economy

**Petra Hospodková, Vladimír Rogalewicz \*** and **Michaela Králíčková**

Department of Biomedical Technology, Faculty of Biomedical Engineering, Czech Technical University in Prague, 272 01 Kladno, Czech Republic
\* Correspondence: rogalewicz@fbmi.cvut.cz

**Abstract:** This study discusses the issue of disposable and reusable medical devices in the context of the circular economy. Its objective is to analyze the opinions of physicians in gynecological practice on the use of disposable and reusable gynecological speculums (GS). A questionnaire survey was carried out in a sample of 206 healthcare facilities in the Czech Republic. In addition to this, the cost of both examination methods was calculated and compared using data provided by a gynecological department of a typical district hospital. The calculations and cost analysis were performed using the life-cycle costing (LCC) method. The economic comparison shows that the use of disposable gynecological speculums is less expensive, having, however, a greater negative impact on the environment. The cost of reusable speculums for 25 examinations per day for 15 years is EUR 51,579, while it is EUR 23,998 for disposable speculums for the same use and time horizon. The questionnaire survey shows that both physicians and patients are more likely to prefer disposable speculums for examination, although without a clear rationale.

**Keywords:** circular economy; gynecological speculum; reusable medical device; disposable medical device; life-cycle costing; LCC; cost analysis





## 1. Introduction

The current economic system is based on the linear economy in which products are disposed of after their use. The aim of the linear economy is to produce, sell, consume and dispose of quickly and cheaply, which should be replaced by long-life products that can be repaired or recycled (Michelini et al. 2017). National and transnational strategies and initiatives are increasingly supporting the health sector in its transition to the circular economy (CE) (European Commission 2022). At the level of the European Union, a comprehensive package dealing with circular economy policy, the so-called circular economy action plan (CEAP), has been adopted since 2015. The plan contains legislative and non-legislative initiatives and represents the largest area of the single market within the circular economy, connecting decentralized public authorities and the business sphere with regional and social funds, and also with the cohesion policy. The EU sees the CE as an opportunity for sustainable economic and environmental strategies. If the CE model is fully implemented, it could have a transformative effect on the economies of all EU member states (Ellen MacArthur Foundation 2020; Dodick and Kauffman 2017).

Healthcare is one of the largest industries in the world. In the EU, the healthcare sector represents 10% of GDP; in the United States even 17.9% of GDP. The highest priority for healthcare providers is above all to provide, safe, quality care, but there should be an effort to minimize and/or recycle medical device waste to save financial resources and protect the environment (Voudrias 2018). Single-use medical devices that burden the environment are currently the most widespread, so an effort can be seen to return to medical devices that can be used repeatedly. The major disadvantage of medical devices for repeated use is disinfection and sterilization to prevent the possible spread of infection

(Moultrie et al. 2015). If (currently single-use) medical devices were to be used repeatedly or recycled, their design would have to be changed. An example of good practice is the Philips company, which refurbishes some medical equipment, such as MRI systems. However, it has been reported that, globally, the healthcare sector lags behind other industries in adopting circular economy practices (Gautam and Sahney 2020). In the healthcare sector, introducing the circular economy into product design is a very challenging process. Designers must consider both legislative requirements and product safety. Reusing medical devices can bring savings to hospitals, but safety, disinfection and sterility regulations must always be followed (van Straten et al. 2021a; Kane et al. 2018). The opinion that disposable medical devices are safer than reusable ones can be abandoned. Single-use medical devices are used much more frequently in practice, although there is no clear evidence that they reduce the risk of infection when providing care. The risks of infection are multifactorial in origin, and the frequency of these infections is so low that studies focusing on specific subjects are difficult to implement because they would require a large number of samples (Macneill et al. 2020).

A typical medical device that raises the question of the choice between disposable and reusable designs is gynecological speculums (GSs). The GS market has experienced major changes in recent years. The annual market growth rate (measured by CAGR) was estimated at 5.09%. The potential growth difference for this market between 2021 and 2026 is estimated at USD 263.23 million. One of the main factors driving the growth of the GS market is the increasing use of disposable GSs (Technavio 2022).

Many authors deal with the issue of assessing economic and environmental impacts of disposable and reusable medical devices (Rizan and Bhutta 2021; Sanchez et al. 2020). A comparative cradle-to-grave life-cycle assessment (LCA) has been recently conducted to determine and analyze the environmental impacts of using disposable acrylic speculums versus reusable stainless steel speculums in a women's university clinic (Rodriguez Morris and Hicks 2022), and an evaluation of the carbon footprint of GSs (a single-use acrylic model and a reusable stainless steel model) has been recently performed (Donahue et al. 2020).

The usual method for calculating costs of a medical device is life-cycle costing (LCC). Sherman et al. (2018) calculated and compared the costs of disposable and reusable laryngoscopes. Ibbotson et al. (2013) performed a similar analysis for disposable and reusable surgical instruments. Research on the cost of GSs using LCC is still limited. The OBP company (OBP 2020) developed a guideline that presents the costs of reusable and single-use GSs. According to their calculations, switching to disposable speculums represents a cost-saving strategy. Considering a practice with an average of 100 pelvic exams a week, they estimated annual savings of over USD 3700.

The discussions around the issue are still hot and it is necessary to consider not only the economic and/or environmental aspects, but also the possibilities of the market and the preferences of gynecologists, which may also be driven by other factors. One of the important trends in the GS market supporting the market expansion are the strategies adopted by the market participants. However, the market growth will be challenged by factors such as a high risk of infection.

The number of treated patients shows a slightly decreasing tendency; in 2020 the field of gynecology and obstetrics recorded 3,179,464 patients, i.e., 586 patients per 1000 women, in the Czech Republic. The total number of treatments/examinations performed in 2020 was 9,109,846, (i.e., 1679 examinations per 1000 women in the population). Despite a slight decrease in the number of examinations, it should be emphasized that most examinations are of a preventive nature and a significant decrease is not expected in the future. In Czechia, there are 1670 gynecological offices, of which 247 operate within larger healthcare facilities (hospitals or polyclinics) (NZIS 2021).

The objective of this study is, first, to find out what physicians prefer in the use of disposable and/or reusable GSs; and, second, to calculate the costs of both GS variants as a pilot estimation based on data from a typical outpatient facility.

## 2. Methodology

### 2.1. Questionnaire Survey

The questionnaire contained study-specific questions developed in discussion between researchers in the research group experienced in teaching and training hospital staff. After this initial step, the questions were discussed by researchers outside the research group (3 gynecologists) to achieve face validity and to obtain opinions on whether the questions were realistic to ask and whether the layout was easy to use. The questionnaire was then revised, and its layout was refined. The final questionnaire included an informed consent and a total of 18 questions (the questionnaire is provided in the Supplementary Materials File). The questionnaire was sent electronically to gynecologist offices operating within hospitals (n = 74) and polyclinics (n = 173) in the Czech Republic, i.e., all 247 larger healthcare centers able to provide a wider range of outpatient services, including gynecology. Gynecologists working in separate independent practices (n = 1423) were not approached. Data collection was carried out from 1 April 2022 to 30 June 2022, and those who did not respond were reminded twice. A total of 206 correctly completed questionnaires were returned (83.5% rate of return).

The statistical analysis was performed using MS Excel. In addition to descriptive statistics, Wilson's score test was used to test preferences between individual types of GSs among physicians on one hand, and among patients on the other.

### 2.2. Cost Calculation

The life-cycle costing method is an economic approach that takes into account all discounted costs of a product, process, or activity during its entire life cycle. This method is not directly related to environmental costs but to costs in general. The traditional LCC is an investment calculation that is used to rank different investment alternatives to help decide the best alternative. LCC is considered a very valuable comparative tool when considering a long-term investment in a product or service (Voelker 1969).

The use of our medical device is defined as: 'Gynecologic speculum is intended for visual examination of the uterine cervix and vagina'. In Table 1 we present the usual structure of cost items (similarly e.g., Estevan and Schaefer 2017; Sherman et al. 2018).

**Table 1.** Costs included in the calculation using the LCC method.

| Cost Category | Cost Item—General View | Reusable GS | Disposable GS |
|---|---|---|---|
| Acquisition costs (A) | Purchase price including discounts | X | X |
| | Shipping and installation costs | X | X |
| | Leasing costs | n/a | n/a |
| | Cost of IT services | n/a | n/a |
| | Adjustment costs for the operation of the equipment | n/a | n/a |
| | Cost of initial training | n/a | n/a |
| | VAT | X | X |
| Cost of operation (O) | Personnel costs | X | n/a |
| | Cost of consumables | X | X |
| | Costs of ongoing employee training | n/a | n/a |
| | Energy and costs for equipment operation | X | n/a |
| | Depreciation | X | n/a |
| | Insurance, taxes, VAT | X | X |
| Service costs (M) | Planned and preventive maintenance costs | X | n/a |
| | Costs associated with cleaning, disinfection and sterilization | X | n/a |
| | VAT | X | n/a |
| Disposal costs (D) | Decommissioning costs | n/a | n/a |
| | Costs of safe disposal | X | X |
| | VAT | X | X |

The table also shows the representation of individual cost categories depending on whether they are reusable or disposable gynecological speculums.

Equation (1) was used to calculate the total *LCC* value of a reusable GS in a particular year. It includes cost categories and a discount factor:

$$LCCr = A + \sum_{t=1}^{n} \frac{(A + O + M + D)}{(1+i)^n} \tag{1}$$

where:

*LCCr* = life-cycle cost for reusable GS
*A* = acquisition costs
*O* = cost of operation
*M* = service costs
*D* = disposal costs
*i* = discount rate
*n* = number of years

Equation (2) was used to calculate the total *LCC* value of a disposable GS in a particular year. It includes cost categories and a discount factor:

$$LCCd = \sum_{t=0}^{n} \frac{(A + 0 + D)}{(1+i)^n} \tag{2}$$

where:

*LCCd* = life-cycle cost for disposable GS
*A* = acquisition cost
*O* = cost of operation
*D* = disposal costs
*i* = discount rate
*n* = number of years

End-of-life costs include safe disposal costs. At the end of their useful lives, both devices are expected to be disposed of as hazardous waste (*Act. No 541/2020 on Waste* 2020). Waste removal (disposal) costs were determined from accounting documents and calculated according to the waste weight. This approach was also used, e.g., by Sanchez et al. (2020).

Since the objective is to compare both variants, operating costs were not calculated as they were assumed to be the same. The real discount rate of 4% (recommended by the European Commission for the year 2021) was used. Since the real discount rate was applied, annual inflation was not included in the net present value calculation because the demand for healthcare is traditionally considered price inelastic (Chakravarty and Debnath 2015).

The crucial question before any LCC calculation is the choice of the time horizon. The first option is the depreciable life representing the time of asset depreciation, while this time parameter depends on the respective national law. Furthermore, we can consider the economic life, the number of years when the asset yield surpasses the costs of its operation and maintenance. The service life represents the number of years that the asset is actually in service. In some cases, the life horizon is fixed to be a predefined number of years, e.g., 5, 10 or 15 years, for example, if the actual economic life and/or service life of the acquisition are uncertain (Solution Matrix Limited 2019). All calculations in this case study are conducted for a period of 5 years (the depreciable lifetime of a reusable GS) and for a period of 15 years (realistic use based on expert opinion and the literature, (Rodriguez Morris and Hicks 2022)). The cost analysis for both GS variants was calculated using data provided by a gynecological outpatient clinic in a single district-type hospital (a typical representative of facilities providing outpatient care in the Czech Republic).

More detailed calculation formulas are given in the Supplementary Materials File.

## 3. Results

### 3.1. Questionnaire Survey

The length of experience of the respondents in the field of gynecology most often ranges from 26 to 35 years (36%), followed by 36 years and more (29%), 16–25 years (25%), 5–15 years (10%) and less than 5 years (<1%).

The representative sample evenly covers all 14 regions of the Czech Republic. In the representation of individual regions, the number of responses varies from ten (Karlovy Vary Region) to 23 (Ústí nad Labem Region).

Almost 61% (127) of the respondents (gynecologists) do not give patients the choice of whether they wish that a disposable or reusable GS is applied, while 39% (79) allow the choice. Fifty-eight percent of gynecologists who allow patients to choose a GS said that patients prefer a disposable speculum. Thirty-six percent of them said that patients prefer a reusable speculum, and six percent claimed that this information cannot be clearly determined. The physicians prefer disposable speculums in 66%, 19% of them do not prefer any particular type of speculum, and 15% prefer reusable speculums.

Table 2 lists the reasons given by the respondents who prefer disposable GSs, and Table 3 lists the reasons given by those who prefer reusable GSs.

**Table 2.** Reasons for using disposable speculums (N = 135).

| Reason for Using Disposable Speculum | Number of Responses | Share |
| --- | --- | --- |
| 1.   No need for disinfection and sterilization | 116 | 86% |
| 2.   Large selection of sizes and designs | 88 | 65% |
| 3.   Better handling and control | 69 | 51% |
| 4.   Greater safety for the patient | 63 | 47% |
| 5.   After arrest, it maintains an open position and allows examination without an assistance of a specialist staff | 62 | 46% |
| 6.   Better visibility of the examined structures | 59 | 44% |
| 7.   It penetrates the vagina better | 37 | 27% |
| 8.   Less costly | 30 | 22% |
| 9.   It is better to take a cytological sample | 28 | 21% |
| 10.   More comfortable for the patient | 11 | 8% |
| 11.   Other | 7 | 5% |

**Table 3.** Reasons for using reusable speculums (N = 31).

| Reason | Number of Responses | Share |
| --- | --- | --- |
| 1.   Better handling and control | 21 | 68% |
| 2.   Greater strength and resistance to breakage | 11 | 36% |
| 3.   Better visibility of the examined structures | 10 | 32% |
| 4.   Large selection of sizes and designs | 10 | 32% |
| 5.   Less costly | 9 | 29% |
| 6.   Environmentally friendly | 7 | 23% |
| 7.   Adapted design for easier cleaning | 2 | 7% |
| 8.   Other | 2 | 7% |

The majority (29%) of the respondents think that reusable speculums are more expensive, 23% believe that the cost of the speculums is about the same, 25% cannot compare their price, and 23% hold that using disposable GSs is more expensive.

The ratio in which gynecologists use the particular types of GSs when performing examinations is presented in Table 4.

**Table 4.** Gynecological speculum usage ratio (N = 206).

| Usage Share during Examination [%] | | Number of Responses | Share of Respondents |
|---|---|---|---|
| Disposable Speculum | Reusable Speculum | | |
| 100 | 0 | 91 | 44% |
| 90 | 10 | 57 | 28% |
| 70 | 30 | 10 | 5% |
| 50 | 50 | 2 | 1% |
| 30 | 70 | 5 | 2% |
| 10 | 90 | 26 | 13% |
| 0 | 100 | 9 | 4% |
| Another ratio | Another ratio | 6 | 3% |

Gynecologists use disposable and reusable GSs with or without locking. Eighty-five percent of them use disposable plastic speculums with locking, 25% use reusable speculums with locking, 15% use reusable speculums without locking, 2% use disposable speculums without locking, and 1% specify use of another type of GS.

Figure 1 shows how gynecologists perceive the benefits for patients when individual types of GSs are applied.

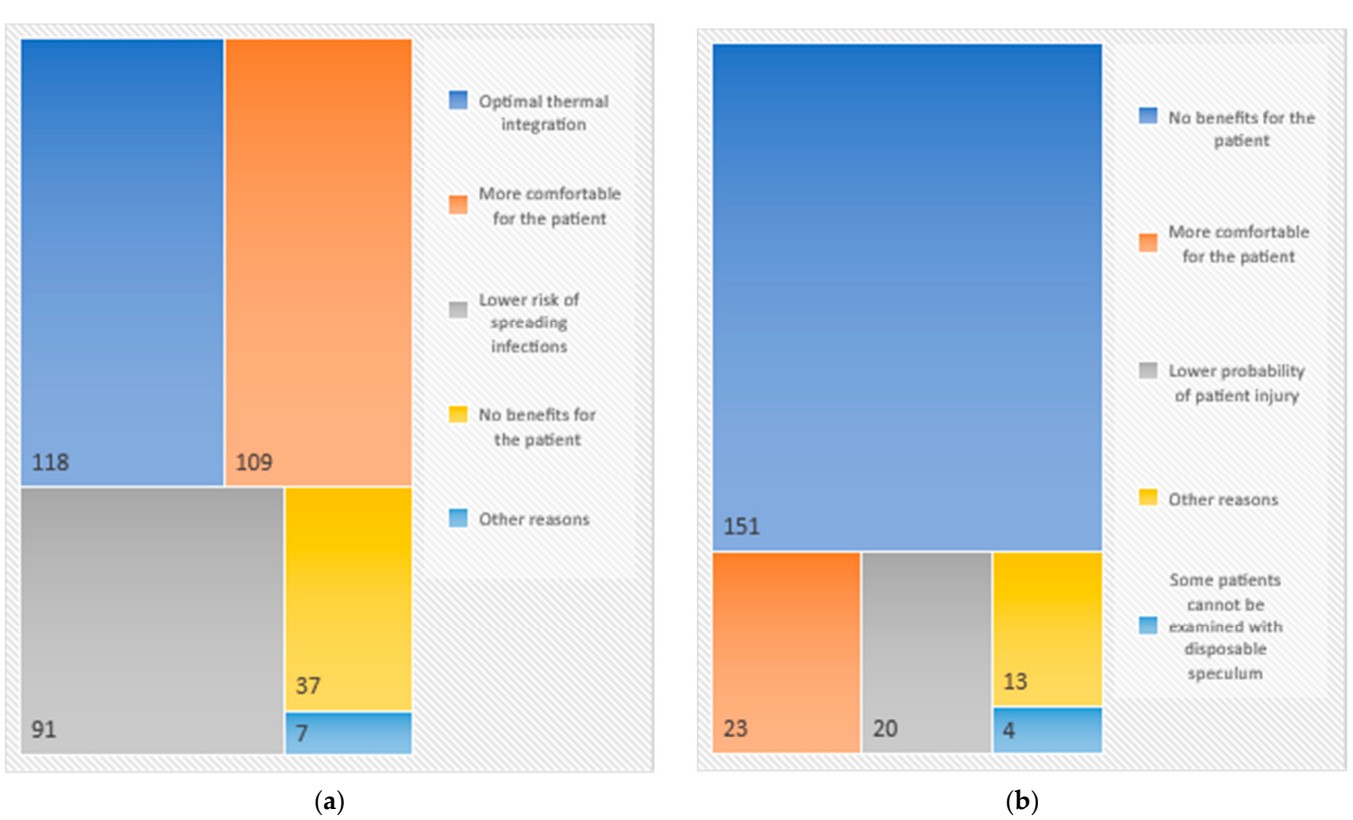

(**a**)　　　　　　　　　　　(**b**)

**Figure 1.** (**a**) Disposable GSs—benefits for patients from the point of view of gynecologists. (**b**) Reusable GSs—benefits for patients from the point of view of gynecologists.

### 3.2. Cost Calculation

Data were provided by a single typical gynecological outpatient clinic in a district-type hospital in the Czech Republic. The calculation assumed the execution of 25 examinations per day for a period of 5 years (the accounting depreciable lifetime of a reusable GS) and for a period of 15 years (realistic life of use). The results are summarized in Table 5.

**Table 5.** Cost of gynecological speculums (5-year and 15-year perspectives) (EUR).

| Cost Category | Cost Item—General View | Reusable GS | Discounted | Disposable GS | Discounted |
|---|---|---|---|---|---|
| Acquisition costs (A) | Purchase price including discounts + shipping and installation costs including VAT | 42.35 | | 9.51 | |
| | Leasing costs | n/a | | n/a | |
| | Cost of IT services | n/a | | n/a | |
| | Adjustment costs for the operation of the equipment | n/a | | n/a | |
| | Cost of initial training | n/a | | n/a | |
| TOTAL (A) (5-year period) | | 1058.65 | 870.14 | 11,411.54 | 9379.45 |
| TOTAL (A) (15-year period) | | 1058.65 | 587.83 | 34,234.62 | 19,009.27 |
| Cost of operation (O) | Personnel costs (specialist staff trained in medical device contamination procedures) | 10.92 | | n/a | |
| | Cost of consumables | 2.90 | | 2.12 | |
| | Costs of ongoing employee training | n/a | | n/a | |
| | Energy and costs for equipment operation | 0.45 | | n/a | |
| | Depreciation (autoclave) | 480.00 | | n/a | |
| | Insurance, taxes, VAT | 2.85 | | 0.21 | |
| TOTAL (O) (5-year period) | | 22,954.20 | 18,866.68 | 2800.05 | 2301.44 |
| TOTAL (O) (15-year period) | | 68,862.60 | 38,236.96 | 8400.15 | 4664.31 |
| Service costs (M) | Planned and preventive maintenance costs | 130.00 | | n/a | |
| | Costs associated with cleaning, disinfection and sterilization | 5.65 | | n/a | |
| | VAT | 28.66 | | n/a | |
| TOTAL (M) (5-year period) | | 7645.35 | 6283.92 | - | - |
| TOTAL (M) (15-year period) | | 22,936.04 | 12,735.57 | - | - |
| Disposal costs (D) | Decommissioning costs | n/a | | n/a | |
| | Costs of safe disposal | 1.88 | | 0.19 | |
| | VAT | 0.41 | | 0.04 | |
| TOTAL (D) (5-year period) | | 11.44 | 9.40 | 285.11 | 234.34 |
| TOTAL (D) (15-year period) | | 34.31 | 19.05 | 585.00 | 324.83 |
| **TOTAL (5-year period)** | | 31,669.64 | **26,030.13** | 14,496.70 | **11,915.23** |
| **TOTAL (15-year period)** | | 92,891.60 | **51,579.41** | 43,219.77 | **23,998.40** |

　　　　The purchase price (including discounts, shipping and VAT) was 42.45 EUR per unit (for reusable GSs) and 9.51 EUR for 25 units of disposable GSs. No additional acquisition costs were identified.

　　　　Personnel costs were determined according to the average hourly wage of a specialist staff trained in medical device decontamination procedures who prepares a disinfectant solution, cleans the gynecological GSs before placing them in the autoclave, and removes the gynecological mirrors from the autoclave after sterilization is complete. All these activities take the specialist staff on average 150 min per day for 25 gynecological mirrors.

　　　　The cost of consumables consists of common disinfectants, but also includes a special disinfectant that is used twice a day to create a disinfectant solution to which 10 L of water is added. After loading into the disinfectant solution, the GSs must be rinsed before sterilization, which requires an average of 20 L of water.

　　　　Energy and cost for equipment operation: the power consumption of the autoclave is 38 kWh and the sterilization time is 50–60 min. The required volume of water for one sterilization cycle is 70 L and the required volume of demineralized water is 8 L.

　　　　Depreciation was calculated based on the purchasing price of an autoclave (data were obtained from the accounting department).

　　　　Service costs were considered for the reusable GSs only. These included the costs of regular safety and technical inspections as well as spare parts.

## 4. Discussion

Although the results of the research indicate that disposable GSs are preferred, it is interesting that only 29% of gynecologists believe that reusable GSs are more expensive and another 23% believe that their price is comparable to disposable GSs. Hence, the decisive factor may not be the price. The research further studied the arguments for the use of individual types of GSs; however, only a small sample of gynecologists mentioned the positive environmental aspect of reusable GSs. A frequently cited factor in favor of disposable GSs was the lower risk of infection. Although the vagina is not a sterile environment, there are cases (such as during an IUD placement) when a sterile environment is desirable (Rodriguez Morris and Hicks 2022). After the use of speculums, sterilization is necessary as an important protection against nosocomial infections, which are later difficult to detect (Southworth 2014). The efficiency of the autoclave is excellent in these cases and if infections occur, it is due to a variety of reasons, mostly non-compliance with the prescribed procedure, ranging from staff, management, equipment or the processes themselves (Panta et al. 2019). Biological waste is dealt with in the Waste Act, Act No. 541/2020 Coll. Hazardous waste is also an object contaminated with biological material marked HP9. The law stipulates special treatment of waste; for example, it must be removed from the medical facility daily or stored in specially designated refrigerators and freezers. In general, waste is treated as hazardous waste. Storage, transport, and disposal are governed by this (the transport of waste is governed by the European directive UN 3291—special provisions VC3 and further by the ADR regulation). The special treatment of waste is due to its potential danger and forces higher costs. The decomposition of the individual arguments that have been presented in favor of reusable GSs can be an interesting stimulus for manufacturers of these products in terms of GSs' design and ergonomics.

Environmental issues in healthcare and the lack of sustainable approaches are becoming a hot topic. The field of gynecology is no exception to this (Hehenkamp and Rudnicki 2022). Healthcare organizations along with other industries, governments and individuals need to join efforts to reduce environmental impacts to make the world more sustainable and healthier (Benedettini 2022).

Although reusable GSs also generate negative environmental impacts during the disinfection and sterilization phase (detergents, sporicidal disinfectants, ethylene oxide, etc. are used), there is evidence (Rizan and Bhutta 2021; Donahue et al. 2020) that the environmental burden from disposable medical devices is significantly higher. One of the issues that needs to be further addressed is monitoring the carbon footprint. van Straten et al. (2021b) proved that single-use medical devices have a 58% higher carbon footprint than reusable medical devices. Some authors (e.g., Rodriguez Morris and Hicks 2022) state that when analyzing environmental impacts, the answer is not clear cut, as the variability of impacts can be very different and always depends on the analyzed product systems, the type of equipment itself and the environmental goals. As a result, studies that analyze reusable versus disposable medical device configurations are becoming more prominent (Byrne et al. 2022; Le et al. 2022).

Many authors deal with redesigning GSs (Wong and Lawton 2021; Urrutia Avila et al. 2021); however, such information reaches gynecologists only very slowly. On the one hand, it is necessary to provide comfort for the patient, while, on the other hand, ensuring quality medical service. Among the most common shortcomings of metal GSs are mentioned great discomfort for patients due to the limited sizing, uncomfortable clicking sounds, uneven pressure, pinching and limited visibility of the monitored object. The results show that satisfactory results in the field of ergonomic properties of medical devices can influence the preferences of physicians when choosing medical devices.

It was shown that the literature on vaginal speculums is generally sparse, and papers on the choice of disposable vs. reusable speculums (including issues of infection transmission) is very rare (Wong and Lawton 2021; SMTL 2000). As was already mentioned, gynecologists often decide between disposable and reusable GSs on the basis of subjective reasons. Tables 2 and 3 may serve as a source of ideas for designing new, enhanced

mirrors. This is an area where principles of the early-stage HTA might be effectively used (Bouttell et al. 2021; IJzerman et al. 2017).

*Limitations of the Study*

In this study, the time considered for sterilization was set at 6 min (this is a parameter that can be variable, since its duration does not depend on the length of the sterilization cycle, which can vary from 30 to 60 min). In the next cost study, it would be appropriate to perform a sensitivity analysis and model this parameter in greater detail.

Given variation in device lifetime (maintenance and attrition factors), it would be useful to add a breakeven analysis.

The study shows some of the problems of this type of research. Although we reached a relatively high return of questionnaires (206 out of 247), there can be some bias caused by the choice to answer that we are not able to analyze. Moreover, answers to some questions were often missing, and hence we had to concentrate to the questions that all the respondents answered. Among the partly omitted information were data necessary for analyzing the effect of the length of practice on the decision between the two types of speculums. As concerns the cost analysis, our intention was just to obtain a rough estimation. Thus, we consider the results of one typical outpatient clinic sufficient to draw attention to the higher cost of reusable devices (in Czechia). If we prefer environmental considerations, it would require a strong response on the part of designers and manufacturers; however, our respondents (clinicians) could not supply their opinion. Researching their views is a challenge for future research.

## 5. Conclusions

Environmental aspects have become very important in designing and using modern technologies, with an emphasis, among others, on replacing disposable devices with reusable variants. We have shown that in the case of gynecological mirrors, both physicians and patients more often prefer disposable speculums, although the reasons are rather emotional or habitual. Physicians often use and patients require what they are used to, and often do not have enough information about all the options. Moreover, the disposable speculums are the cost-effective variant at present. Safety reasons play an important role, as they affect the maintenance costs of reusable devices significantly. For designers and manufacturers, this presents a challenge to find solutions that make the reusable variant more attractive, just as environmental considerations need to be better promoted.

**Supplementary Materials:** The following supporting information can be downloaded at: https://www.mdpi.com/article/10.3390/economies11020070/s1: The questionnaire incl. the informed consent (in Czech and in English), and more detailed formulas for calculating the LCC.

**Author Contributions:** Conceptualization, P.H. and V.R.; methodology, P.H. and V.R.; software, M.K.; validation, P.H., V.R. and M.K.; formal analysis, V.R.; investigation, P.H.; resources, M.K.; data curation, M.K. and V.R.; writing—original draft preparation, P.H.; writing—review and editing, V.R.; visualization, M.K.; supervision, V.R.; project administration, P.H.; funding acquisition, P.H. All authors have read and agreed to the published version of the manuscript.

**Funding:** This research was supported by an internal grant from Czech Technical University in Prague No SGS22/132/OHK4/2T/17.

**Institutional Review Board Statement:** Not applicable.

**Informed Consent Statement:** Not applicable.

**Data Availability Statement:** The data are available upon reasonable request to the first author. The data are available partly in the Czech language.

**Acknowledgments:** The authors would like to thank Radek Poláček, for valuable consultations during completing and editing this paper.

**Conflicts of Interest:** The authors declare no conflict of interest. The funder had no role in the design of the study; in the collection, analyses, or interpretation of data; in the writing of the manuscript; or in the decision to publish the results.

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
