# Peer review of "Gynecological Speculums in the Context of the Circular Economy"

_economies, doi:10.3390/economies11020070_

Round 1
Reviewer 1 Report
Manuscript: economies-2142760
Title: Gynecological Speculums in the Context of the Circular Economy Submitted to section: Labour, Health, and Education
Date: 03-01-2023
General comment: I welcome these kind of studies investigating the impact of disposable versus reusable medical devices. To my opinion this study provides an added value to the discussion regarding the use of disposable versus reusable medical devices. However, it could elaborate more on details such as costs. The discussion is whether cost savings is the only determining factor to acquire certain types of devices. The discussion incorporating the design aspects of sustainable medical devices is a good aspect of this study.
Comment 1: Introduction: National and transational strategies and initiatives are increasingly supporting the health sector in its transition to the circular economy (CE). This is a statement and could be cited. I suggest to add a source (e.g. European Green Deal: https://commission.europa.eu/strategy-and-policy/priorities-2019-2024/european-green-deal_en or publications on circular healthcare economy citing these).
Comment 2: I agree with the authors that there is no clear evidence that disposable devices reduce the risk of infection. I leave it up to the authors however, mentioning a sentence that the cleaning, disinfection and sterilisation in the EU (or developed countries) of reusable medical devices through a validated central sterilization and services process (CSSD), ensures a high level of infection prevention. There is no evidence that using disposables decreases the risk of infection significantly when having a thorough CSSD process in place.
Comment 3: The abbreviation GS is described half way through the introduction. I suggest to write that out or ad between brackets behind earlier description on this page.
Comment 4: I recommend to add a table. A table besides the current costs of the LCC method, with the structure of the costs containing line level specifications of the costs incurred between reusable and disposable. Also the Supplemental is not clear what the (average) original costs are per device. An enumeration of the costs gives the reader a better understanding of the built-up of the cost price (i.e. purchase costs, costs of cleaning, disinfection, sterilisation, disposal costs (ranging in many EU countries from 1 – 1,85 Euro per kg for hazardous waste). Please also include VAT as this is a cost price increasing factor. The VAT can often not be reclaimed by hospitals. That refers to the VAT of both the original purchase price of the device as well as the VAT on the waste disposal costs.
Comment 5: Costs calculations in a hospitals should be approached from a holistic perspective, meaning taking all costs into consideration. With some regularity, cost comparisons in healthcare do not include all actual costs as these are either not transparent or difficult to perceive. A note of this aspect could be added to the discussion.
Comment 6: The discussion could be improved. I would suggest to add a comment in the discussion regarding the value of conducting a Life Cycle Assessment or referring to a study which conducted a LCA on medical devices (e.g. the study on LCA on reprocessing face masks, 2021 or the publications on the environmental impacts of medical devices). In this context also the environmental benefits of a reusable (in terms of costs) devices could be stipulated as weighing factor next to the costs. Climate change impacts lead to increased costs on the longer term. Please consider to describe the importance of a LCA in this context.
Comment 7: Please add some sentences on the increased costs of raw materials (oil & gas) which have their effect on disposable products in particular. Next to the cost increase of raw materials including metal-based materials, also the costs of energy have impact on the purchasing costs of medical devices. Furthermore, increase of CO2 taxes as well as the Corporate Social Responsibility Directive (CSRD), to be introduced in 2024, are expected to have consequences for suppliers of medical devices. The CSRD states that from 2024 more and more companies will be obliged to report on their impact on people and on the climate. This will be of influence on the supply of disposables. https://finance.ec.europa.eu/capital-markets-union-and-financial-markets/company-reporting-and-auditing/company-reporting/corporate-sustainability-reporting_en
Comment 8: Conclusion: ‘Disposable things’. I suggest to change the word ‘thing’ to product or device.
Comment 9: The area of medical devices is vague. Do you mean the development of medical devices? And delete the word ‘this’. The sentence then flows better.
Comment 10: The sentence in line 238 (We showed…..) does not read smoothly. Please rewrite.
Comment 11: To my idea the Conclusions do not reflect the content of the manuscript. Please add more relevant content from the study.
Comment 12: : Supplemental. The questionnaire should be translated into English in order for the readers to comprehend.

Reviewer 2 Report
I'm delighted to see this topic of inquiry and the authors are to be commended for their work. That said, I do have several comments to improve their presentation.
· Overall this is well written, however it is clearly not by a native English speaker. It would improve the strength of the paper to have someone fluent in English medial writing to edit the paper.
Methods
· The authors need to clearly explain how their survey was created, by whom, whether it was validated (i.e., do questions mean what the researchers intend/are they properly interpreted?)
· As the survey is in Czech, I am unable to interpret it. It would be preferable to have this translated into English.
· The authors need to give clearer descriptions in how many potential respondents there were in the 173 gyn clinics and 74 hospitals. 206 completed questionnaires out of how many potential? What is the percentage completed?
· Table 1: is this reproduced from Estevan et al? most of these categories do NOT apply to simple medical devices such as vaginal specula, and should be deleted to prevent any confusion by the readers.
· The authors note that specula are treated as hazardous waste. Really? This is not standard. They can be safely treated as regular (municipal) solid waste.
· It should be more explicit that costs were taken from a single hospital, and the type and size of that hospital should be described. (Is it a representative cost model?)
Results
· In the abstract the authors note annual savings of EUR 1851 but in the results section they note a savings of USD3700. Firstly, they should be consistent in units. Secondly, they need to be clear what year their calculation was performed (these numbers are presently very different than expected.)
· Results should be presented in terms of absolute results in addition to relative results, in the prose sections.
· Figure 1 is difficult to read and should be converted to two bar graphs.
· The authors base their calculations on 25 examinations a day for 5 years, stating this is the “accounting lifetime” and 15 years noting this is the “real lifetime.” These terms are not clear and need to be defined. Stainless surgical steel in reality can last for many years, and some simple medical equipment does not even have a stated lifetime so long as they are maintained (e.g. surgical steel screws replaced on occasion). Where do these numbers come from? While the authors note more information can be had upon request, it is simple enough to work backwards from their table; instead of challenging the reader to do so, it would be far more helpful to the reader if they could please provide per use costs and do a break-even analysis—how many uses to achieve cost parity? The numbers 1) are difficult to believe and 2) do not match the results in the abstract and those in the results as pointed out, above.
Discussion
· The authors should attempt at least some review of the infection control literature. Is there any evidence to suggest that specula cause infections? If they have, were there extenuating circumstances?
· The authors should discuss other potential shortcomings of disposable devices, e.g., that they are too flexible, making it more difficult to achieve the desired view and tissue samples. Is there anything in the literature on this?
· The authors should discuss some known solutions that improve patient comfort of reusable steel, e.g. running it under warm tap water for a few seconds—common practice (noting, of course that these costs were not considered in their study.)
Round 2
Reviewer 2 Report
the authors are to be commended for a high quality, scientific paper. I do find it odd that their results are different than expected, based on from other papers.
1) Can the authors confirm that a nurse performs this duty? It is typical to have specialist staff trained in medical device decontamination procedures, and it is a lower pay-scale than nursing staff. Their approach makes reusables look less favorable
2) can the authors describe how they determined how long it takes to clean 25 specula? The note an average of 6 minutes per device, whereas other studies with similar devices (McGain and also Sherman) of laryngoscope (similar class/complexity as the speculum) noted two minutes per device ); these had similar findings and are from two different countries. Their approach makes reusables seem less favorable
3) the authors note that waste disposal class is biohazardous. I can't find any citation to support this claim. This actually increases the costs of the disposables relative to reusables.
4) Related to (3) By the Spaulding Classification system, these devices are considered intermediate risk, requiring a minimum of high level disinfection (not even sterilization.). While there may be select cases where sterility would be desirable (and certainly not achievable), e.g., IUD placement, it is not necessary for the vast majority of cases. Thus, their estimates of reusable costs are over-estimated, and this should be noted.
5) given variation in device lifetime (maintenance and attrition factors), it would be useful to add a breakeven analysis. At what number of uses would cost parity be achieved?
6) harm to human health from climate change and environmental degradation results in more demand for healthcare. Thus, the authors should either note this as a limitation, or even better, than can add the social cost of carbon as one accounting approach
